# Roll-to-roll manufacturing of flexible acetone sensors

**Ya-Ching Yu[1], Nicholas Glassmaker[2], Ana M. Ulloa[1], Benson Kunhung Tsai[1], Amit Barui[1], Haiyan Wang[1], Lia Stanciu**  [1,3,4]*

**1** School of Materials Engineering, Purdue University, West Lafayette, Indiana, United States of America,
**2** Birck Nanotechnology Center, Purdue University, West Lafayette, Indiana, United States of America,
**3** School of Biomedical Engineering, Purdue University, West Lafayette, Indiana, United States of America,
**4** Bindley Bioscience Center, Purdue University, West Lafayette, Indiana, United States of America

* lstanciu@purdue.edu

## Abstract

A fully roll-to-roll (R2R) manufactured conductometric sensor has been developed for acetone detection. It features flexible, screen-printed silver electrodes modified with a MoS2 and single-walled carbon nanotube (SWCNT) nanocomposite. The electro-spraying process allowed for the simultaneous fabrication of over 100 electrodes. The optimized electro-sprayed electrodes showed a consistent average resistance of 99.63 Ω (SD=8.10), ensuring uniform and reliable sensor performance for volatile organic compounds (VOCs) detection. SEM and TEM analyses revealed that the porous 3D $MoS_2$-SWCNT network enhances gas adsorption and sensor performance. To improve acetone selectivity, electrodes were functionalized with tetra-fluorohydroquinone (TFQ), which binds with acetone and facilitates charge transfer, reducing holes in p-type SWCNT and increasing resistance. Sensitivity was further optimized by adjusting film deposition parameters, with a thinner $MoS_2$ layer (~30 nm) improving the response to 20% acetone by 120%. Stability tests confirmed 90% signal retention after 30 days in nitrogen. This study proposed the first platform of a $MoS_2$-SWCNT nanocomposite in R2R manufacturing for flexible acetone sensors. The approach provides a scalable solution, bridging lab-scale research with industrial production and advancing practical VOC detection applications.

## 1. Introduction

Since the start of the Industrial Revolution, the release of volatile organic compounds (VOCs) into the environment has significantly risen due to activities like petroleum refining, car exhaust emissions, indoor furniture manufacturing, and waste management [1]. VOCs are a class of organic chemicals that easily evaporate into the atmosphere, where a majority become hazardous pollutants with wide-ranging implications [2]. Their presence poses a significant threat to human health, environmental safety

**Data availability statement:** All relevant data underlying the findings of this study are available in the Purdue University Research Repository (PURR) at https://purr.purdue.edu/projects/acetonesensors. The dataset includes raw numerical values corresponding to all figures in the manuscript: Dataset_Fig1-3.xlsx – Response (%) versus VOC type and TFQ functionalization, calibration curves. Dataset_Fig4-6.xlsx – Response (%) versus acetone concentration (ppm) and time (s), and elemental compositions (atomic %). Dataset_Fig7-8.xlsx – Response (%) versus configuration number, sensor stability (G/G$_o$ %), and AFM height profiles (height nm vs position μm). These files contain the minimal data needed to reproduce the results and conclusions reported in the article. Figs 1 and 4 are schematic representations and contain no numerical data. Data were digitized from the author's laboratory records to provide long-term accessibility and compliance with the PLOS ONE Data Availability Policy.

**Funding:** Funding for this project was provided by the SMART Film Consortium in BNC at Purdue University and the US Department of Agriculture, Agricultural Research Service, under Agreement ARS-CFSE funding (no. 59-8072-6-001), project [no. 8072-42000-077-00D] and National Science Foundation (CBET award no 2127756). The funders had no role in study design, data collection and analysis, decision to publish, or preparation of the manuscript.

**Competing interests:** The authors have declared that no competing interests exist.

[3,4]. Some VOCs are particularly concerning due to their potential carcinogenic effects [4,5]. These compounds, when inhaled in substantial amounts, can lead to severe health issues ranging from respiratory disorders to neurological damage and, in some cases, cancer [5]. Alarmingly, the concentration of VOCs indoors can be up to ten times higher than outdoor levels, subjecting individuals to continuous exposure in everyday environments such as homes and workplaces [6,7]. On the other hand, while VOCs are often considered hazardous due to their potential environmental and health impacts, they also present unique opportunities for non-invasive diagnostics. Their presence and concentration in biological systems, such as breath, can serve as critical biomarkers for diseases, like lung, adenomas, and colorectal cancer, and can thus present opportunities for early disease detection and personalized healthcare solutions [8–10].

Among the various VOCs, acetone stands out as one of the most widely used industrial solvents [11]. Although considered less harmful than other VOCs, exposure to acetone can exacerbate the toxicity of other substances, like acetonitrile [12]. Acute exposure to acetone, even in relatively low concentrations, can result in symptoms such as headaches, dizziness, and skin irritation [12]. On the other hand, acetone and other VOCs, such as methanol, ethanol, and isopropanol, also serve as health biomarkers that reflect metabolic processes and disease states [13]. Elevated acetone levels in breath, for example, indicate diabetic ketoacidosis [14]. Detecting VOCs through non-invasive methods, like breath analysis, offers a convenient way to monitor and diagnose conditions like diabetes, cancer, and liver disorders [15]. This makes the detection of VOC interesting from the point of view of industrial environmental safety, but also relevant to low-cost health monitoring, such as for an alternative to traditional blood glucose testing for early disease diagnosis and management [15].

Transition metal dichalcogenides (TMDs), such as molybdenum disulfide ($MoS_2$), emerged as promising materials for gas sensing due to their unique two-dimensional structure, high surface area, and tunable electronic properties [16–18]. However, the integration of $MoS_2$ into scalable sensor platforms while maintaining high sensing performance, remains a challenge [19–21]. The increased awareness of the importance of VOC detection has spurred the advancement of new sensing technologies, particularly those based on two-dimensional (2D) materials, including $MoS_2$ [22–25]. These materials, owing to their exceptionally high surface-to-volume ratio, provide large active surface areas that increase molecular adsorption, which is a key requirement for sensor applications [22,23,25]. The large surface area allows such sensors to achieve higher sensitivity and faster response times than those achieved by conventional materials. TMDs in particular offer the advantage of semiconducting properties, mechanical durability, and environmental stability [26]. Among these, $MoS_2$ has attracted significant interest due to its adaptable structural phases, electronic properties, and availability of active sites for surface functionalization [21,27,28]. These attributes make $MoS_2$ a superior material for VOC sensing when compared to alternatives like graphene, which lacks a band gap, and MXenes, which are prone to environmental degradation over time [29,30].

Although MoS$_2$ is a promising 2D semiconductor material for conductometric sensors, it inherently lacks sufficient electrical conductivity and selectivity for efficient gas or VOC sensing. To overcome these limitations, integrating conductive materials and functionalization for selective detection are necessary steps towards the design of an effective VOC sensing system. According to the literature, semiconductor materials are often combined with noble metal nanoparticles (such as Au, Ag, and Pt) or carbon-based materials like graphene and carbon nanotubes in an effort to increase their electrical conductivity. In Chen's study, single-walled carbon nanotubes (SWCNTs) were combined with MoS$_2$ to significantly boost the performance of VOC sensors [31]. These hybrid materials create interconnected networks that improve charge carrier mobility, reduce electrical resistance, improve sensitivity, and minimize signal noise. In this report, this synergy rendered the MoS$_2$-SWCNT composites, when coated with Cu(I)-pincer complexes for specificity, highly effective for detecting VOCs like ethylene [31]. However, the use of traditional drop-casting methods for depositing MoS$_2$-SWCNT mixtures is likely to lead to inconsistent film thicknesses, which limits scalability and large-scale production readiness of the platform.

As discussed above, the relevance of VOC detection is rapidly expanding across diverse fields, including environmental monitoring, early diagnosis of diseases, agricultural pathogen detection, and food quality control. Among the many approaches available, chemiresistive sensors, which operate by altering resistance in the presence of target gases, emerged as one of the most simple and effective approaches for VOC detection [32,33]. These sensors provide high sensitivity, stability, rapid response, as well as cost-effectiveness and ease of fabrication [32,34]. The emergence of flexible electronics has further advanced the capabilities of such chemiresistive sensors. Flexible electronics allow the integration of sensor technologies onto lightweight, adaptable substrates such as plastics or paper [35]. For example, flexible sensors can be embedded into wearable devices for continuous health monitoring or applied in environmental sensing devices that require a high degree of versatility [36,37]. Combining flexible electronics with chemiresistive sensors improves the adaptability of the sensors and aligns with the growing trend toward miniaturization and portability in electronic devices. The advantages are even greater when these flexible sensors are produced using R2R manufacturing techniques. R2R manufacturing allows for scalable, cost-efficient sensor production with consistent performance and high throughput. In the study by Ulloa et al., a fully R2R manufactured electrochemical sensor was developed for detecting nitroaromatic organophosphorus pesticides (NOPPs) [38]. By combining graphene's conductivity and zirconia's electrocatalytic activity, the sensor demonstrated high sensitivity (down to 0.2 ppm), group selectivity, and stability over 30 days, showcasing the potential for scalable, reliable pesticide detection [38].

In this study, we introduce the integration of chemiresistive sensor technology with R2R fabrication, towards a versatile platform for the large-scale production of flexible, high-performance VOC sensors. The use of MoS$_2$-SWCNT hybrid nanocomposites leverages the sensor's sensitivity and stability, while functionalization with TFQ enables selective acetone detection. The R2R process efficiently and effectively deposits MoS$_2$-SWCNT materials onto flexible, screen-printed silver electrodes, with high consistency of properties, in a fashion that overcomes the limitations of laboratory-level sensor fabrication methods.

## 2. Experimental section

### Preparation of MoS$_2$ suspension

15.89 mg of MoS$_2$ powder with sizes <2 μm and 99% purity from Sigma-Aldrich was suspended in 3 mL of 45 vt% ethanol/water solution. The mixture was sonicated for 2 h in an ice bath by using high power mode in a Branson CPX (2800 H) to exfoliate the MoS$_2$ nanoflakes. SWCNT was purchased from SuperPureTubes, NanoIntegris with 0.025 wt% resuspended in water for the following R2R spraying process.

### Manufacturing process

An array of conductive interdigitated electrodes (IDEs) was fabricated by screen-printing silver paste (DuPont 5025) on printed flexible polyethylene terephthalate (PET) substrates (DuPont Teijin Films ST505, 0.005-inch-thick × 14 inches

wide × 500 ft. long roll). IDEs were printed and dried in a semicontinuous R2R process using a machine designed and built by Kinzel Printing Systems GmbH (Bielefeld, Germany) (see Fig A1 in S1 File). IDEs ultimately used for this study had overall dimensions of 0.96 × 1.4 cm and 600 µm spacing and were printed in a repeating fashion with a spacing of 4 mm (see Fig A2(a) in S1 File). After printing and drying were completed, a R2R slitting machine was used to create two 7-inch-wide films with various repeating electrodes printed on each film. The reduction in film width was necessary to make the film compatible with the Maxwell machine used for electro-spraying.

## Roll-to-roll MoS$_2$ and SWCNT coating

The MoS$_2$ suspension and SWCNT suspension were electro-sprayed via a R2R process on silver electrodes using the custom-designed and built R2R Maxwell machine located at Purdue University's Birck Nanotechnology Center (see Fig A3(a) in S1 File). The Maxwell system was used to unwind, tension, and transport the film through a temperature and humidity-controlled cabinet at 20 °C and 42% RH. The IDEs were connected by copper tape for grounding. Inside the cabinet, the MoS$_2$ suspension was electro-sprayed from the nozzle aligned at a height of 30 mm over top of the sensing end squares of the electrodes, whereas the substrate was transported past the nozzles at a speed of 400 mm/min. The tips of the needles were biased to a voltage of 16 kV relative to the machine ground, which was connected to the grounding strip (and connected electrodes) utilizing a wire compressed against the passing strip, as shown in (see Fig A3(a) in S1 File). The MoS$_2$ suspension flow rate was measured as 0.193 mL/min. With these settings, the width of the sprayed area was approximately 50 mm so that the sensing ends of the electrodes were fully covered. The illustration is shown in (see Fig A2(b) in in S1 File). After spraying, the electrodes traveled through a second stage of the Maxwell system that consisted of hot plates at 100 °C, where complete drying was ensured. After that, the SWCNT solution was sprayed on top of the MoS$_2$ film by the same procedures. The tips of the needles were biased to a voltage of 16 kV with a 0.415 mL/min flow rate and the substrate was transported at a speed of 400 mm/min. Carbon nanotubes served as channels electrically connecting the MoS$_2$ nanosheets, which enhanced the charge carrier transfer. Through alternate spraying of these two materials, the layer-by-layer spraying resulted in a nanohybrid structure MoS$_2$-SWCNT nanocomposite exhibiting good sensing performance to VOCs.

## TFQ functionalization

Dispersions of MoS$_2$-SWCNT mixtures were deposited onto IDEs by electro-spraying with final resistances in a range of 90–110 Ω. TFQ solution was synthesized with 0.025 mM in acetone and modified according to the previous study [39]. MoS$_2$-SWCNT films were then functionalized by casting 1, 2, 5, 10, 20, 30, and 50 µL drops of TFQ and dried in the hood as an ultrathin film. The as-prepared sensors could be stored in a desiccator for the following measurement.

## Analytical instrumentation

The surface morphology and microstructure of MoS$_2$-SWCNT nanocomposite films were investigated using a Hitachi S-4800 for SEM analysis. The atomic force microscopy (AFM) image was performed by a commercial scanning probe microscope from Bruker (Dimension Icon). The AFM images were taken under Scanasyst mode by using the SCANASYS-AIR tip from Bruker. Bright-field transmission electron microscopy (TEM), specifically the Tecnai G2 20 (operating at 200 kV with a LaB6 filament, Oxford Instruments), was utilized to take images of the MoS$_2$-SWCNT nanohybrid structure.

## VOC sensing measurements

The performance of the MoS$_2$-SWCNT sensors in the detection of various VOCs was measured with a homemade gas-sensing system. (see Fig A3(b) in S1 File) Briefly, the sensors were placed in a sensing chamber equipped with a

gas inlet and outlet. Mass flow controllers (Brooks 5850E) were used to control the concentrations of VOC analytes by adjusting the flow rates of the VOC analytes and dilution gas (dry air), with a total flow rate fixed at 500 mL/min. The bubbler containing the analyte was controlled at a given temperature to maintain a stable vapor pressure. The target gas and purging gas (nitrogen) were exposed for 5 and 10 min for each cycle of gas-sensing testing to achieve a steady-state measurement, respectively. An input voltage of 1V was applied to three electrodes of the sensors inside the measuring chamber (see Fig 2(d)), and the real-time current signals were acquired at room temperature using a PalmSens4 in Potentiostat.

The response of a $MoS_2$-based sensor upon the adsorption of a VOC analyte is defined by the following equation

$$Response\ (\%) = \frac{\Delta I}{I_0} \times 100 = \frac{|I_g - I_0|}{I_0} \times 100$$

where $I_g$ and $I_0$ represent the current values of the sensor in the presence of VOC analyte and nitrogen, respectively.

## 3. Results and discussion

Fig 1 illustrates the R2R fabrication process for conductometric acetone sensors based on $MoS_2$-SWCNT films coated with TFQ complexes. The process begins with the preparation of a diluted aqueous suspension of exfoliated $MoS_2$ nanoflakes. Using electro-spraying techniques, this suspension is deposited onto the silver IDEs printed on flexible PET substrates, forming the base $MoS_2$ layer. After allowing the $MoS_2$ layer to dry, a similar electro-spraying process is used to deposit single-walled carbon nanotubes (SWCNTs) onto the $MoS_2$ layer. This alternating layer-by-layer deposition resulted in the formation of a $MoS_2$-SWCNT nanocomposite on the IDEs.

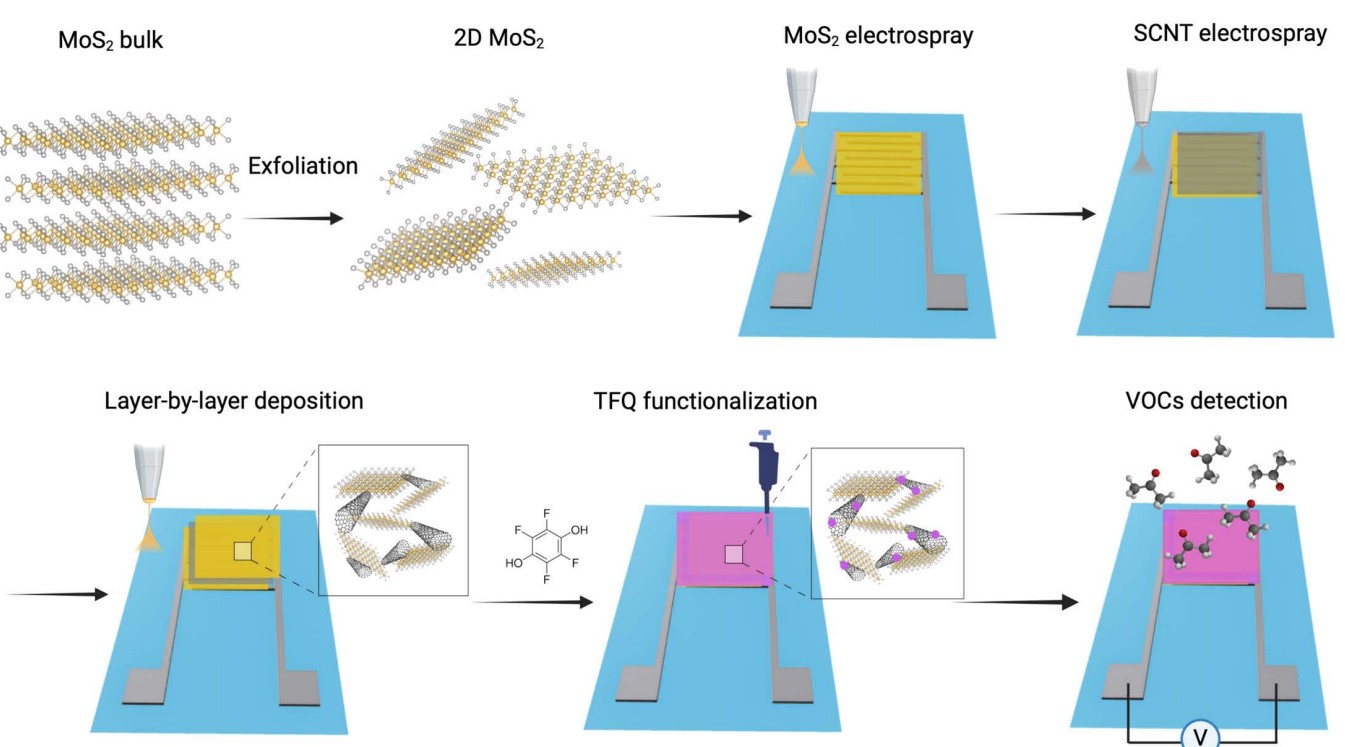

**Fig 1. Illustration of $MoS_2$ exfoliation and R2R electro-spraying process of $MoS_2$-SWCNT sensors for VOC detection.**

The surface morphology and microstructure of the MoS$_2$-SWCNT network were analyzed using bright field SEM imaging, as shown in Fig 2(a,b). The exfoliated MoS$_2$ nanoflakes with feature sizes in the range of several hundred nanometers were evenly dispersed, creating a continuous three-dimensional (3D) network connected by SWCNT bundles. This porous 3D structure offered a high specific surface area, which significantly increased gas adsorption and improved sensing performance. Further microstructural insights were obtained from TEM analysis (2(c)), where the carbon nanotubes were observed to form conductive pathways between the MoS$_2$ nanosheets, improving charge carrier transfer. This structure greatly increased both the sensitivity and conductivity of the sensor, as shown by the red arrows indicating these nanotube connections.

To evaluate their performance for conductometric VOC detection, interdigitated electrodes (IDEs) coated with MoS$_2$-SWCNT networks were tested in initial experiments. As shown in Fig 2a, three IDEs were placed inside a flow cell at room temperature, and exposed to a constant gas flow rate of 0.5 L/min. Electrodes were connected to a potentiostat to monitor changes in current, initially being exposed to nitrogen for 10 minutes, followed by a 5-minute exposure to 20 vt% of various VOC analytes mixing with 80 vt% of nitrogen. The observed change in current reflects a temporary alteration in the sensor's electronic states caused by charge transfer interactions between the MoS$_2$-SWCNT composite and adsorbed VOC molecules. For instance, acetone, an electron-withdrawing molecule, captures electrons from the conduction band of the n-type MoS$_2$ upon adsorption, reducing carrier density and leading to a decrease in current [40]. When the MoS$_2$-SWCNT film is exposed to VOC analytes, the molecules are adsorbed onto its nanoporous network structure and with-draw electrons from the film, further decreasing the current. The results demonstrated that the nanoporous structure (Fig 2a,b) effectively captures VOC analytes, although the sensor's specificity was found to be limited, as shown in 2 (e).

Before initiating the electro-spraying R2R process with the final optimized settings, a series of preliminary experimental trials were conducted to fine-tune the parameters for a uniform and reproducible MoS$_2$-SWCNT coating. Stainless steel

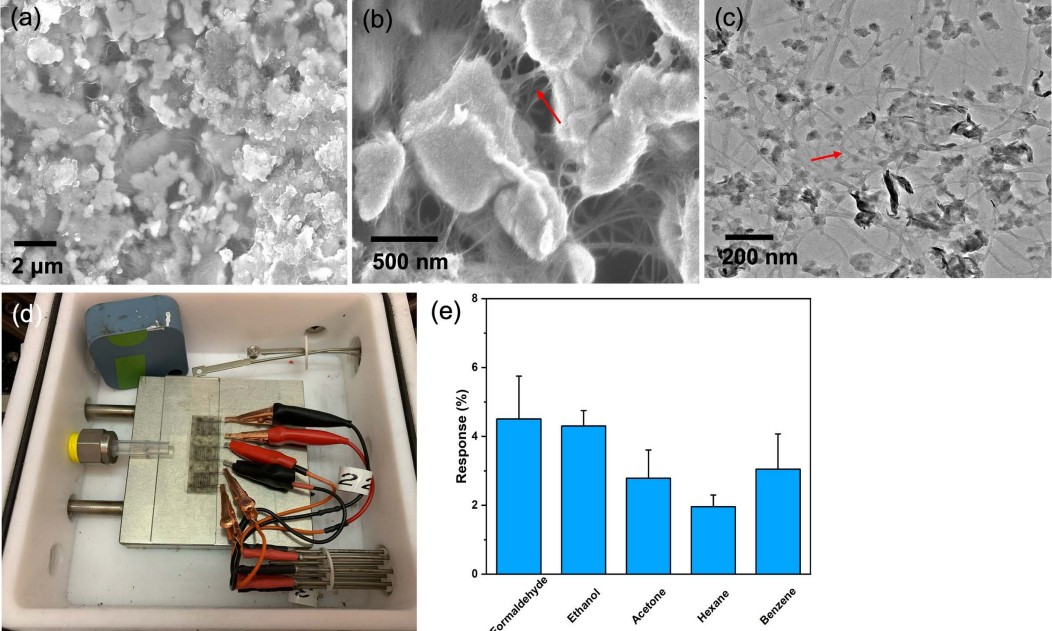

**Fig 2. Electron microscopy images of the composite materials, along with images of the experimental set up, and VOC sensor response data.** (a) SEM images showed the MoS$_2$-SWCNT composite structure. (b) MoS$_2$ nanosheets and SWCNTs created a nanoporous network that improves gas adsorption efficiency. (c) TEM images revealed that carbon nanotubes act as conductive channels linking the MoS$_2$ nanosheets, improving charge carrier transfer, which significantly increased sensitivity and conductivity. (d) Three electrodes are set in the gas-sensing chamber. (e) The electrodes produced using R2R methods demonstrated effective VOC detection capabilities.

needles, each with an inner diameter of 337 µm, were utilized for the electro-spraying process. To prevent the suspension from flowing out of the nozzles when the voltage was turned off, the material supply container for each nozzle was maintained under a slight vacuum (~10 mmHg). This vacuum level helped control the flow of the suspension, and further fine adjustments were made by introducing air through electronic pressure regulators. These adjustments reduced the vacuum level slightly, allowing for a steady, controlled, and uniform spray from each nozzle.

The operating settings depend on each other, and optimizing the process depends on the specific applications. The shape and size of dried particles are influenced by processing factors such as distance from tip to the ground, flow rate, and applied voltage [41]. First, the needle tip height is relayed to droplet evaporation time and size. Heights ranging from 30 mm to 60 mm above the electrodes were tested, with 30 mm showing the most uniform spray pattern and optimal droplet size. This height was selected to ensure the desired coverage across the electrode surface. Second, the applied voltage is an important parameter for preparing monodisperse particles because different voltages result in changes to particle morphology. In literature, increasing the voltage up to 20 kV was found to reduce the particle diameter [42]. Based on the experimental data (summarized in Table 1), the deposition of $MoS_2$ suspension was tested from 13 kV to 17 kV. In dripping mode, applying low voltage causes the liquid to form large, stretched droplets that often exceed the nozzle diameter. When the voltage was gradually increased, the highly charged droplets were ejected in a steady and continuous liquid stream. As for the pressure, it represented the force to push out the $MoS_2$ suspension and tested from 3 to 7 psi. When the pressure increased, the higher flow rate led to poor solvent evaporation and a larger deposition width, as shown in The deposition width for $MoS_2$ and SWCNT suspensions refers to the area covered when these suspensions are electro-sprayed onto the PET substrate. The narrow sample deposition width helps capture all particles effectively and keeps them within the electrode area. The same process optimized the deposition of SWCNT suspension to achieve good coating quality. After that, a voltage of 16 kV and a pressure of 3 psi were determined to be the ideal settings for both $MoS_2$ and SWCNT suspensions with appropriate deposition width and uniform thin-film coating on the IDEs.

Table 1. Results of the optimization of the electro-spraying parameters through experimental trials.

| Suspension Type | Pressure (psi) | Voltage (kV) | E-spraying consistency | Deposition width (cm) |
|---|---|---|---|---|
| $MoS_2$ | 3 | 13 | Drop | N/A |
| $MoS_2$ | 3 | 14 | Uniform spray | 4 |
| $MoS_2$ | 3 | 15 | Uniform spray | 3.5 |
| $MoS_2$ | 3 | 16 | Uniform spray | 3.5 |
| $MoS_2$ | 3 | 17 | Spray too fast | N/A |
| $MoS_2$ | 5 | 13 | Drop | N/A |
| $MoS_2$ | 5 | 14 | Uniform spray | 6.5 |
| $MoS_2$ | 5 | 15 | Uniform spray | 6 |
| $MoS_2$ | 5 | 16 | Uniform spray | 5.5 |
| $MoS_2$ | 5 | 17 | Spray too fast | N/A |
| $MoS_2$ | 7 | 13 | Drop | N/A |
| $MoS_2$ | 7 | 14 | Uniform spray | 6.5 |
| $MoS_2$ | 7 | 15 | Uniform spray | 6.5 |
| $MoS_2$ | 7 | 16 | Uniform spray | 6 |
| $MoS_2$ | 7 | 17 | Spray too fast | N/A |
| SCNT | 3 | 16 | Uniform spray | 5 |
| SCNT | 3 | 17 | Uniform spray | 5.5 |
| SCNT | 3 | 18 | Uniform spray | 5 |
| SCNT | 3 | 19 | Uniform spray | 5 |
| SCNT | 3 | 20 | N/A | N/A |

As a result, this optimized electro-spraying process made the continuous R2R fabrication of more than 100 electrodes at the same time possible. The electrodes showed an average resistance of 99.63 Ω (SD = 8.10), which proved that these settings provided reliable and uniform spraying, enabling the production of high-quality coatings. Then, the uniform coating and reproducible process guaranteed the reliability of the sensors in this study and laid a strong foundation for the subsequent VOC detection tests. The optimization of the electro-spraying parameters leveraged the development of high-performance sensors with good sensitivity to acetone and signal stability, which could be translated into the development of sensors for other VOCs (Table 1).

Based on prior research, $MoS_2$'s properties make it an appropriate candidate material for signal transduction in similar sensors [19–21,31,43,44]. However, as shown in Chen et al.'s study, $MoS_2$ alone, when applied to IDEs, generated electrodes with very high resistance and sensor responses with high levels of noise, limiting this material's practical application in conductometric sensing [31]. This demonstrates that the use of conductive materials, such as SWCNTs to create conductive electrodes is necessary for adequate sensor performance. Previous studies have demonstrated that metallic SWCNTs significantly improved charge mobility and work synergistically with semiconducting materials like $MoS_2$ in a percolation network [45,46]. This interaction increased electron transport, reducing the effective channel length and overcoming the noise limitations observed with $MoS_2$ alone.

To further improve the conductometric detection capabilities of the sensors towards acetone, IDEs coated with $MoS_2$-SWCNT networks were functionalized with varying volumes of TFQ. The mechanism by which TFQ improves selectivity toward acetone involves both chemical interactions and electronic effects. TFQ molecules are known to interact non-covalently with carbon-based nanomaterials such as single-walled carbon nanotubes (SWCNTs) via π–π stacking and hydrophilic interactions. When TFQ is deposited on the $MoS_2$–SWCNT composite film, it forms a uniform thin layer that modulates the surface chemistry of the sensor.

Acetone has a relatively high dipole moment (2.69 D) compared to other tested VOCs such as ethanol or hexane, which strengthens its interaction with the TFQ layer (Fig 3(h)). This strong interaction leads to a more pronounced charge transfer between acetone and the TFQ-functionalized sensing surface. Specifically, acetone, acting as an

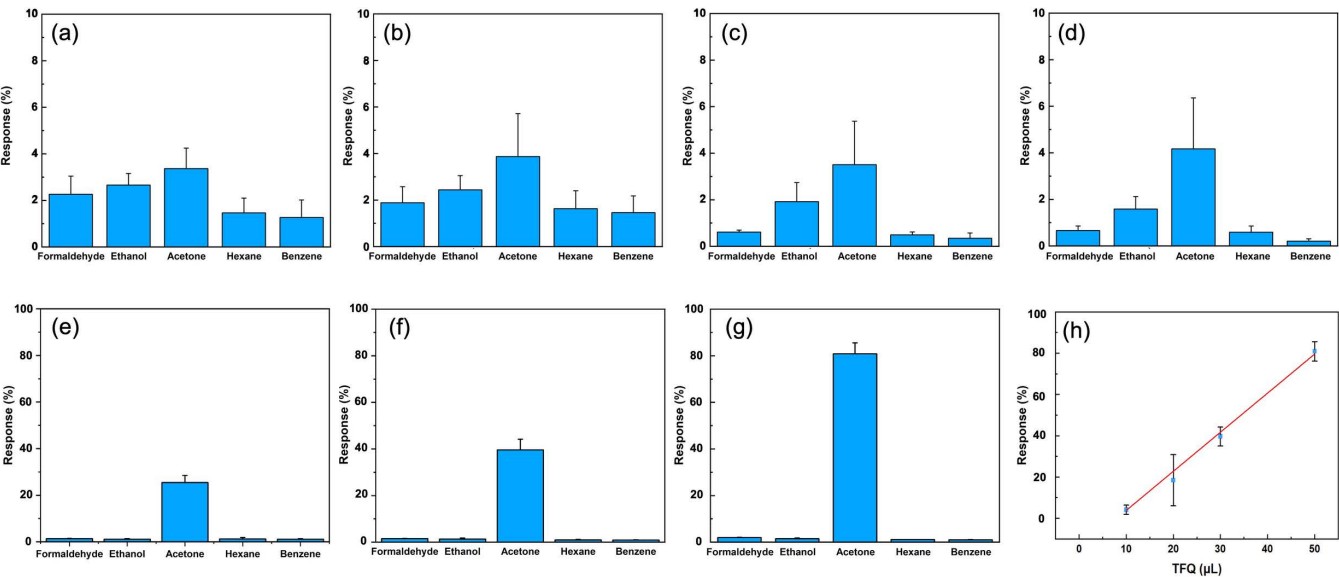

**Fig 3. The response of the R2R fabricated acetone sensors upon electrode functionalization with TFQ.** (a-g) The response of functionalized R2R sensors by adding 1, 2, 5, 10, 20, 30, 50 µL of TFQ respectively. (f) The response of acetone is linearly increased with respect to TFQ. The correlation coefficient is 0.998.

electron-withdrawing molecule, facilitates the transfer of electrons from TFQ to the p-type SWCNTs. This results in a reduction in hole concentration and an increase in the sensor's resistance, which increases the selectivity and signal strength for acetone.

In literature, carbon-based nanomaterials can be non-covalently modified by TFQ molecules through π-π stacking or hydrophilic interactions, which preserves SWCNT's original sp$^2$ structure and high electron conductivity and sensitivity [47]. As shown in Fig 3(a–g), MoS$_2$-SWCNT sensors were treated with TFQ volumes of 1, 2, 5, 10, 20, 30, and 50 µL, allowing the solvent to evaporate in the hood at room temperature. The sensing experiments were conducted using a homemade gas-sensing system, as described in our previous work [31]. In each experiment, three replicates were performed for verifying repeatability. The sensors' responses were tested against five different VOCs, each at a 20 vt% concentration. Initially, when only small amounts of TFQ were applied to the electrodes, there was minimal to no discernible response to acetone, indicating insufficient interaction at low concentrations (Fig 3a–d). However, as the amount of TFQ increased (Fig 3e–g), the sensors displayed a steadily stronger response, confirming that TFQ facilitated acetone detection. This improvement might be due to defect-free TFQ functionalization, which increased both response and selectivity [48]. Since the hydroxyl groups in TFQ bind weakly with acetone molecules, it enables easy charge transfer between SWCNT and acetone. Electrons move from TFQ to SWCNT, reducing holes in the p-type SWCNT and increasing its resistance.

The relationship between TFQ volume and the sensor's response to acetone was found to be linear, with a correlation coefficient of 0.998 (see Fig 3h), which demonstrates that higher TFQ concentrations proportionally increase acetone sensitivity. This suggests that the introduction of TFQ significantly improves the sensor's ability to detect acetone at various concentrations. In Fig 4a and d, SEM images showed the morphology before and after depositing TFQ on top of the MoS$_2$-SWCNT nanohybrid structure. In addition, EDS mapping images (see Fig 4b and e) also validated the increasing atomic ratio of carbon, oxygen, and fluorine resulting from TFQ deposition (also see quantification results in Fig 4f).

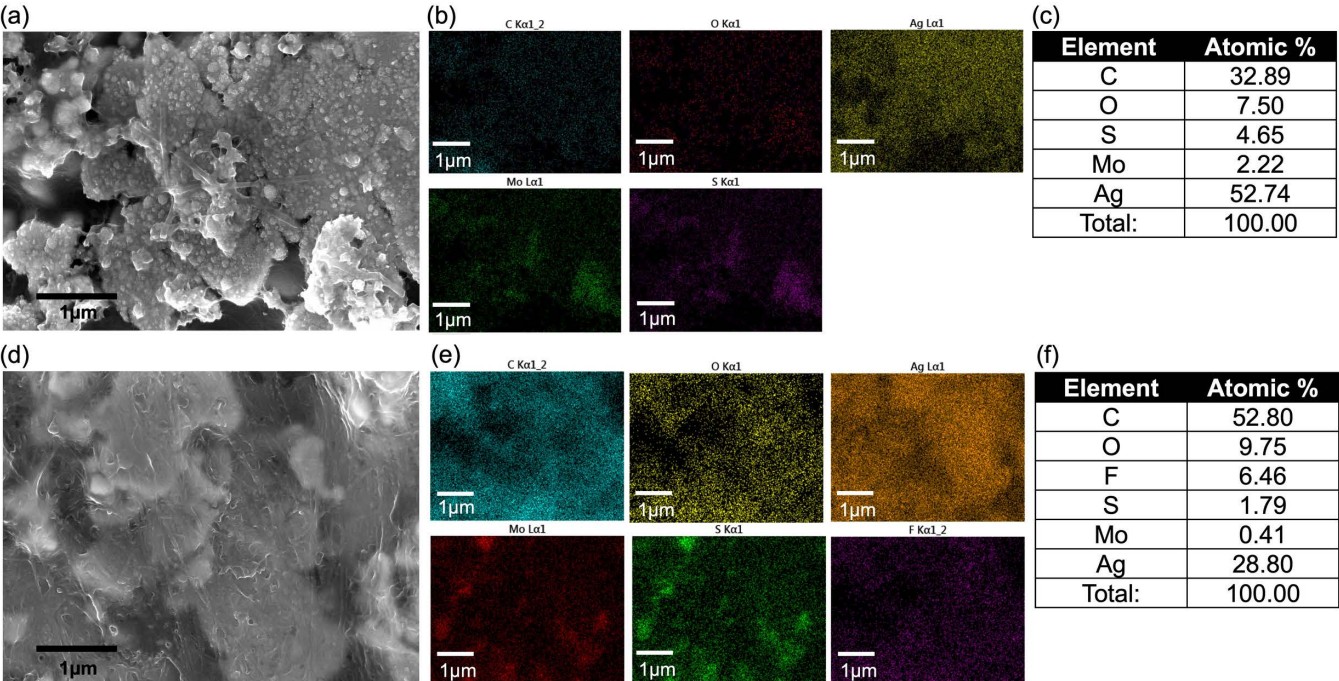

(c)

| Element | Atomic % |
|---------|----------|
| C | 32.89 |
| O | 7.50 |
| S | 4.65 |
| Mo | 2.22 |
| Ag | 52.74 |
| Total: | 100.00 |

(f)

| Element | Atomic % |
|---------|----------|
| C | 52.80 |
| O | 9.75 |
| F | 6.46 |
| S | 1.79 |
| Mo | 0.41 |
| Ag | 28.80 |
| Total: | 100.00 |

**Fig 4. Scanning electron microscopy and elemental mapping results and the effect of TFQ functionalization on composite microstructure and composition.** (a) and (d) SEM image of MoS$_2$-SWCNTs composite without and with TFQ functionalization. (b) and (e) Elemental mapping of images (a) and (d). (c) and (f) Quantification results in atomic ratio.

The dynamic response of the sensor was characterized by monitoring current vs. time during acetone exposure and recovery. As shown in Fig 5, the sensor requires approximately 2188 s (36 minutes) to reach 90% of its full response upon introduction of 50 ppm acetone, and similarly ~2188 s to recover to 90% of baseline after acetone is removed. These response and recovery times reflect the kinetics of analyte adsorption/desorption in the $MoS_2$ SWCNT-TFQ sensing layer. While these response and recovery times are relatively long, we note that this is likely due to the thick film morphology and strong analyte adsorption with TFQ functionalization. Further improvements in response speed may be achieved by optimizing film morphology or operating temperature.

Fig 6 illustrates the sensor response as a function of acetone concentration. The response increases approximately linearly with concentration between 5 and 50 ppm acetone. From the linear regression, the sensor's sensitivity is calculated to be 0.42% per ppm. Many other chemiresistive sensors for VOCs, especially acetone, report sensitivities in the range of 0.1% to 1.0% per ppm, depending on the material system and fabrication method. Unlike many high-sensitivity sensors that are drop-cast or microfabricated, this one is roll-to-roll manufactured, meaning 0.42%/ppm is excellent given the scalability and reproducibility. The limit of detection (LOD) is estimated to be ~2.3 ppm, based on the 3σ noise criterion. Specifically, the LOD was calculated as the concentration corresponding to three times the standard deviation of the baseline noise). These results demonstrate the sensor's relevance for practical applications such as diabetic breath monitoring, where breath acetone can range from 1.8–20 ppm depending on metabolic state. Our sensor isn't just sensitive but it's also selective, thanks to TFQ functionalization. This reduces false positives in complex VOC environments, adding to the value of the measured sensitivity.

The sensor responses at 10 ppm acetone varied by a standard deviation of ~0.2% (absolute resistance change) among devices, which corresponds to roughly 7% of the mean response, which is a reasonable reproducibility. Error bars in Fig 6 represent the standard deviation of the response from three separately fabricated sensors at each concentration. The low variation (average relative standard deviation ~6.5%) demonstrates good reproducibility of the R2R fabrication process. This suggests that our roll-to-roll printed sensors have consistent performance across different devices.

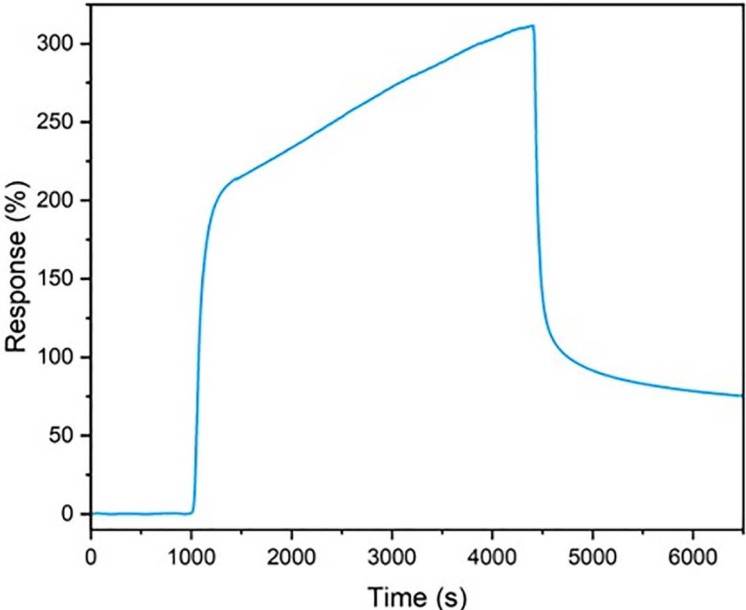

**Fig 5.  The acetone sensor current vs. time during acetone exposure and recovery towards 20 vol% acetone.**

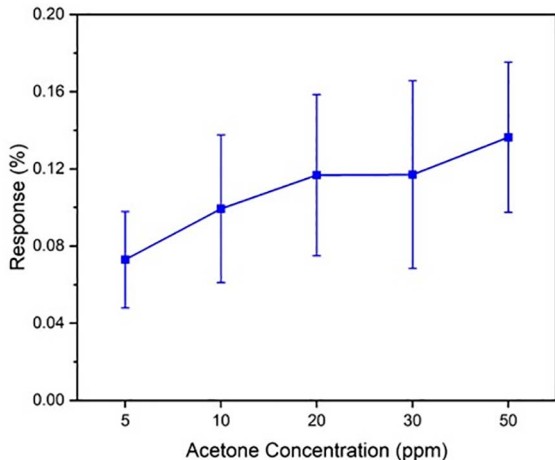

**Fig 6. Acetone calibration curve for the roll-to-roll printed sensor.** The sensor's response ($\Delta R/R_0\%$) is plotted against acetone concentration (5–50 ppm). Error bars represent the standard deviation from three separate sensors, indicating device-to-device variation.

To optimize the R2R layer deposition process for improved sensitivity, multiple spraying sequences were tested to identify the most effective configuration, as shown in Fig 7(a). In this setup, the yellow layers correspond to the printing of $MoS_2$, and the gray layers represent the deposition of SWCNT. The $MoS_2$ layers (~80 nm thickness) were printed first to achieve a uniform and expansive surface area across the substrate. A uniform electrode coating is necessary for a good interaction between the sensor's surface and target molecules. Following this, the SWCNT layer (~70 nm thickness) was applied on top to improve the system's overall electrical conductivity. By alternating these material layers, a three-dimensional (3D) nanoporous architecture was created, which is highly advantageous for VOC detection as it increases the active surface area to facilitate VOC molecule diffusion. The R2R schematics and photographs of results are shown (see Fig A4 in S1 File).

In the fifth tested configuration, the $MoS_2$ layer was intentionally made thinner by increasing the speed of the PET substrate during the printing process to 2.5 times. This faster substrate speed resulted in a reduced $MoS_2$ coating thickness (~30 nm thickness), optimizing material usage while maintaining sensitivity. To counterbalance the thinner coating, three successive $MoS_2$ layers were printed to create uniform and continuous coverage. This specific configuration proved to be the most effective, delivering the highest signal response, with a 120% increase in sensitivity when exposed to 20 vt % acetone, as shown in Fig 7(b).

For assessing the long-term stability of the sensors, both TFQ-functionalized $MoS_2$-SWCNT sensors and $MoS_2$-SWCNT sensors (used as a control) were subjected to storage in a nitrogen atmosphere in a desiccator. Over a month, these sensors were periodically tested to evaluate their response to acetone (with N = 3 measurements). As time progressed, the performance of the $MoS_2$-SWCNT control sensors declined, retaining only 80% of their original response by the end of the month. In contrast, the TFQ-functionalized sensors showed superior stability, maintaining nearly 90% of their performance after the same duration, as illustrated in Fig 7(c). The extended stability of TFQ-functionalized $MoS_2$-SWCNT sensors might come from the TFQ film effectively covering the active edge sites on $MoS_2$ sheets, helping maintain sensitivity over long storage time. The results demonstrate the durability of TFQ-functionalized $MoS_2$-SWCNT sensors, which makes them promising candidates for real-world applications.

In an additional experiment, AFM was used to measure the thickness of the $MoS_2$-SWCNT film on the electrode. Fig 8(a) shows the 3D morphology between the PET substrate to the $MoS_2$-SWCNT film deposited on the electrode.

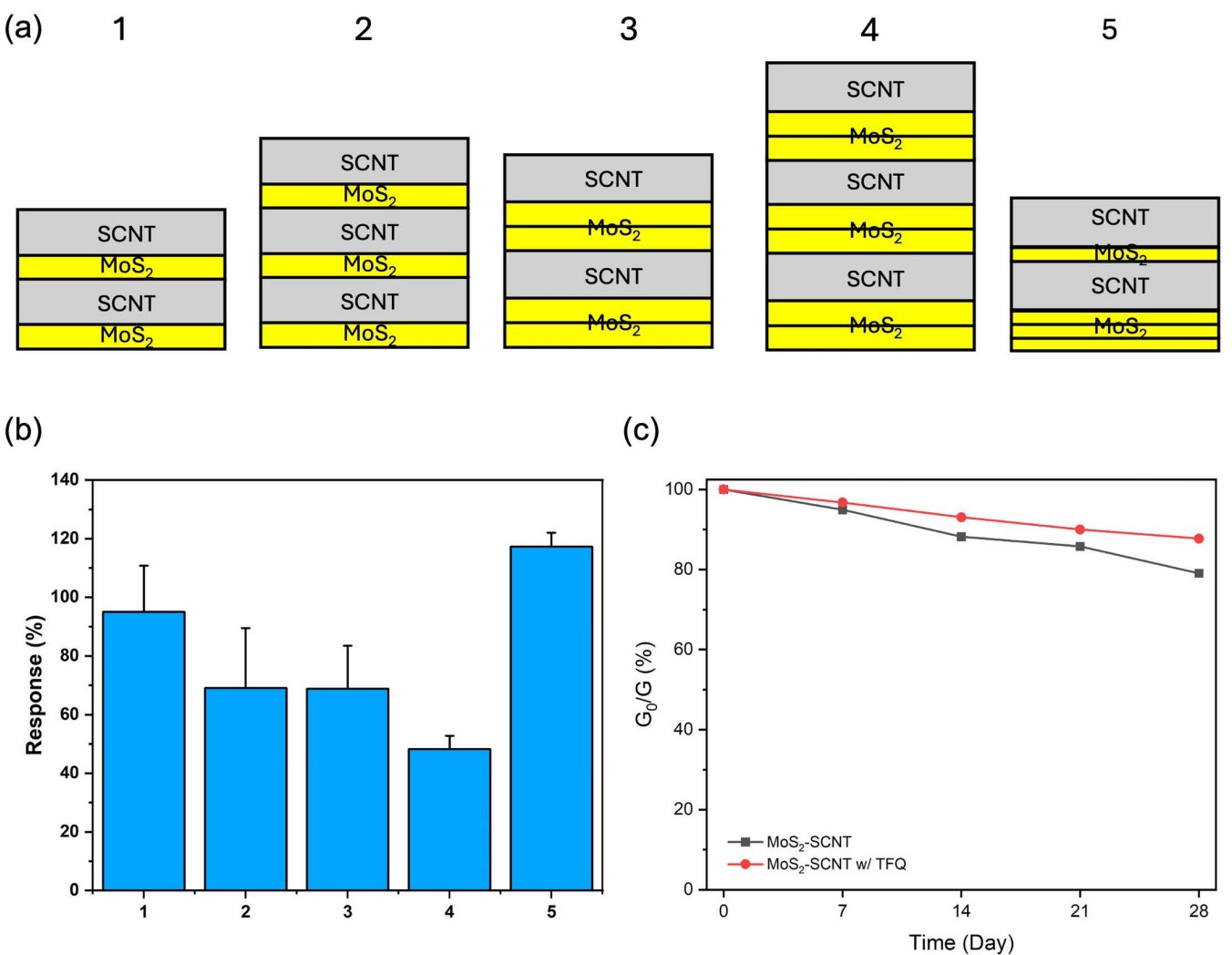

**Fig 7. Optimization of electrode R2R processing parameters and effect on VOC sensing performance and stability.** (a) Optimization of R2R printing layer arrangements. (b) Response of five different printing arrangements with 20 vt% acetone detection. (c) The stability test of MoS$_2$-SWCNT sensors functionalized with TFQ and MoS$_2$-SWCNT sensors.

The thickness of the MoS$_2$-SWCNT film was approximately 300 nm from the cross-section profiles (Fig 8c), which was indicated by lines in the AFM topography image, as shown in Fig 8b.

## 4. Conclusions

This study presents a R2R manufactured conductometric sensor designed for acetone detection, marking a significant step toward scalable, real-world sensor applications. The sensor employs flexible, screen-printed silver electrodes modified with a MoS$_2$-SWCNT nanocomposite, creating a highly porous 3D network that increased VOC adsorption diffusion. These optimized sensors showed an average resistance of 99.63 Ω (SD = 8.10) with uniform and reliable coating. The electro-sprayed MoS$_2$-SWCNT layers functionalized with increasing volumes of TFQ significantly improved acetone detection, showing a linear correlation (R$^2$ = 0.998) between TFQ volume and conductometric response. Additionally, adjusting the R2R spraying process for a thinner MoS$_2$ layer (~30 nm) improved the acetone signal by 120%.

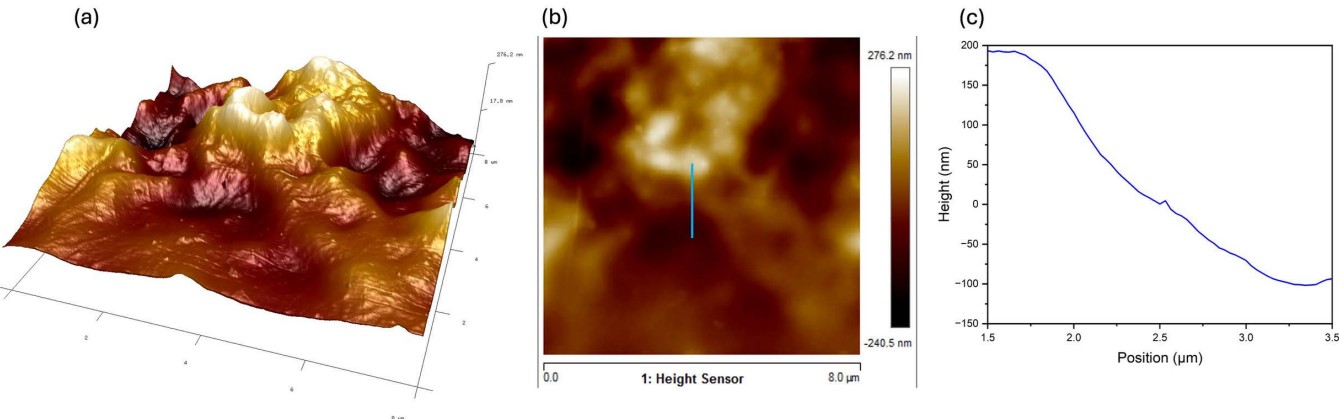

**Fig 8. Atomic force microscopy analysis of VOC sensor surface characteristics.** (a) 3D morphology of MoS$_2$-SWCNTs film on IDEs electrodes. (b) AFM topography image and (c) corresponding cross-section profiles indicated by lines in the AFM images (b).

Stability tests confirmed that the sensors retained 90% acetone response over one month. The R2R manufactured and TFQ-functionalized sensors reported herein have the potential to contribute to the future development of other scalable, easy-to-manufacture, and cost-effective sensing devices for applications in environmental monitoring, industrial safety, and healthcare. While this study focused on demonstrating scalable fabrication and improved selectivity toward acetone via TFQ functionalization, the stability and sensitivity of the sensors across 10–50 ppm concentration range was also assessed, with a LOD found to be approximately 2.3 ppm. Future work will address an improvement of sensitivity across a broader range of acetone concentrations, including trace amounts. The current sensor design and readout system are optimized for detecting moderate to high acetone concentrations. Ongoing efforts are aimed at adapting the system for low-concentration detection through signal amplification and calibration, which will support potential applications in non-invasive diagnostics and environmental monitoring.

## Supporting information

**S1 File. Fig A1(a) Kinzel roll-to-roll (R2R) screen printer (b) Screen-printing silver paste was applied to the patterned screen shown here to print silver electrodes.** (c) IDEs were printed and dried in a semicontinuous R2R process (transport belts for intermittent substrate motion are shown). Fig A2(a) Electrode dimensions. (b) Screen-printed silver electrodes and sensor layout for R2R Printing. Fig A3(a) Electrospraying setup for fabricating MoS$_2$ and SCNT films. The electrosprayed electrodes were around 70 cm long. (b) Homemade gas-sensing system for VOCs measurements. Fig A4 Schematics and photographs of results of various R2R printing layer arrangements.
(DOCX)

## Author contributions

**Conceptualization:** Ya-Ching Yu, Lia Stanciu.

**Data curation:** Lia Stanciu.

**Investigation:** Ya-Ching Yu, Ana M. Ulloa.

**Methodology:** Ya-Ching Yu, Nicholas Glassmaker, Benson Kunhung Tsai, Ana M. Ulloa, Amit Barui.

**Resources:** Nicholas Glassmaker, Amit Barui, Haiyan Wang, Lia Stanciu.

**Supervision:** Ana M. Ulloa, Lia Stanciu.

**Validation:** Ya-Ching Yu.

**Writing – original draft:** Ya-Ching Yu.

**Writing – review & editing:** Lia Stanciu.

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
