## [Decision Letter · Decision Letter 0]

13 May 2025

Dear Dr. Stanciu,

Thank you for submitting your manuscript to PLOS ONE. After careful consideration, we feel that it has merit but does not fully meet PLOS ONE’s publication criteria as it currently stands. Therefore, we invite you to submit a revised version of the manuscript that addresses the points raised during the review process.

**ACADEMIC EDITOR: **

I have completed my evaluation of your manuscript. The reviewers recommend reconsideration of your manuscript following minor revision. I invite you to resubmit your manuscript after addressing the comments below.

We look forward to receiving your revised manuscript.

Kind regards,

Yadvendra Singh, Ph.D.

Academic Editor

PLOS ONE

“Funding for this project was provided NSD CBET EAGER Award No. 2348775 and the US Department of Agriculture, Agricultural Research Service, under Agreement ARS-CFSE funding (no. 59-8072-6-001), project [no. 8072-42000-077-00D].”

“Funding for this project was provided NSF CBET EAGER Award No. 2348775 and the US Department of Agriculture, Agricultural Research Service, under Agreement ARS-CFSE funding (no. 59-8072-6-001), project [no. 8072-42000-077-00D].”

“Funding for this project was provided NSD CBET EAGER Award No. 2348775 and the US Department of Agriculture, Agricultural Research Service, under Agreement ARS-CFSE funding (no. 59-8072-6-001), project [no. 8072-42000-077-00D].”

Reviewers' comments:

Reviewer's Responses to Questions

**Comments to the Author**

1. Is the manuscript technically sound, and do the data support the conclusions?

Reviewer #1: Yes

Reviewer #2: Yes

Reviewer #3: Partly

2. Has the statistical analysis been performed appropriately and rigorously?

Reviewer #1: No

Reviewer #2: Yes

Reviewer #3: Yes

3. Have the authors made all data underlying the findings in their manuscript fully available?

Reviewer #1: Yes

Reviewer #2: Yes

Reviewer #3: Yes

4. Is the manuscript presented in an intelligible fashion and written in standard English?

Reviewer #1: Yes

Reviewer #2: Yes

Reviewer #3: Yes

Reviewer #1: This manuscript presents an original and technically significant study on the roll-to-roll (R2R) manufacturing of flexible conductometric acetone sensors using MoS₂-SCNT nanocomposites functionalized with tetrafluorohydroquinone (TFQ). The topic is timely and relevant, addressing scalable fabrication for VOC detection. While the experimental work is well-executed and clearly described, there are several critical aspects that require clarification or further development to meet the standards of scientific rigor and completeness:

1. The manuscript lacks a quantification of the limit of detection for acetone, which is essential to evaluate the practical applicability of the sensor, especially in clinical or environmental contexts. Authors should include dose–response curves across a range of acetone concentrations and determine LOD values.

2. While several VOCs were tested, there is no quantitative discussion on sensor selectivity or cross-sensitivity against typical interfering gases (e.g., humidity, CO₂, NH₃). These aspects should be experimentally addressed or at least discussed.

3. The response of the sensor is evaluated as a function of TFQ volume but not as a function of acetone concentration. Authors are encouraged to include calibration curves to show sensitivity and linearity across varying acetone concentrations.

4. Multiple references to figures in the text are broken (e.g., “Error! Reference source not found.” on pages 8 and 12). This must be corrected throughout the manuscript before publication, as it disrupts the readability and undermines the presentation of results.

5. Error bars are presented in figures without specification (e.g., standard deviation or standard error). Moreover, there is no indication of statistical testing to support the observed differences. Authors should apply appropriate statistical analyses (e.g., ANOVA, t-tests) and report p-values where relevant.

6. The authors mention TFQ as a functionalizing agent for enhancing acetone selectivity, but do not provide a clear scientific rationale for choosing TFQ over other established functional groups. A comparative discussion based on binding affinity, stability, or prior precedent in VOC sensing literature should be included to justify this choice.

Reviewer #2: While the manuscript presents good results on the roll-to-roll fabrication and functionalization of flexible acetone sensors, I have a few important concerns that should be addressed to strengthen the work:

1. Response time is a critical performance metric for real-time sensing applications. I recommend that the authors include time-resolved current response plots showing both the rise and decay phases of the signal, along with quantitative response and recovery times (e.g., time to reach 90% of the steady-state signal).

2. Several figure citations in the text appear as “Error! Reference source not found". Also, I could not view the full captions of some figures. Please ensure that all figures are complete, properly labeled, and include full, descriptive captions.

3. The manuscript uses "SCNT" for single-walled carbon nanotubes, but the correct and standard abbreviation is "SWCNT". Please revise this throughout the manuscript.

4. Please report the limit of detection (LOD) for acetone, as this is essential to evaluate the sensor’s sensitivity and applicability for low-concentration environments such as breath analysis or indoor air monitoring.

Reviewer #3: The manuscript titled "Roll-to-Roll Manufacturing of Flexible Acetone Sensors" presents a promising approach for scalable fabrication of MoS₂-CNT-based acetone sensors via roll-to-roll (R2R) processing. The enhanced selectivity achieved through TFQ functionalization is particularly noteworthy. However, the manuscript requires improvements to address the following key points, which are essential for a comprehensive evaluation and comparison of the sensing performance

The authors should provide analysis of how the sensor response varies with different concentrations of acetone. A calibration curve with multiple concentration points would help assess the sensor’s sensitivity over a range of concentration, and limit of detection

The information on the temporal behaviour of the sensor, including response time and recovery time need to be addressed.

To demonstrate reproducibility and reliability of the R2R manufacturing process, statistical data on the variation in sensor response across multiple devices should be included

**Do you want your identity to be public for this peer review?** For information about this choice, including consent withdrawal, please see our Privacy Policy

Reviewer #1: No

Reviewer #2: No

Reviewer #3: No

---

## [Author Response · Author response to Decision Letter 1]

11 Jun 2025

Response to Reviewers Questions

What is the mechanism by which TFQ improves the selectivity of acetone sensors?

Answer. We thank the reviewer for this important question. The mechanism by which tetrafluorohydroquinone (TFQ) improves selectivity toward acetone involves both chemical interactions and electronic effects. TFQ molecules are known to interact non-covalently with carbon-based nanomaterials such as single-walled carbon nanotubes (SCNTs) via π–π stacking and hydrophilic interactions. When TFQ is deposited on the MoS₂–SCNT composite film, it forms a uniform thin layer that modulates the surface chemistry of the sensor.

Acetone has a relatively high dipole moment (2.69 D) compared to other tested VOCs such as ethanol or hexane, which strengthens its interaction with the TFQ layer (see Fig. 3(h) in the manuscript). This strong interaction leads to a more pronounced charge transfer between acetone and the TFQ-functionalized sensing surface. Specifically, acetone, acting as an electron-withdrawing molecule, facilitates the transfer of electrons from TFQ to the p-type SCNTs. This results in a reduction in hole concentration and an increase in the sensor's resistance, which increases the selectivity and signal strength for acetone.

We added text in the revised manuscript to describe this mechanism.

For the experiment section in page 6, the authors mention that MoS2-SCNT films were then functionalized by casting 1, 2, 5, 10, 20, 30, and 50 μL drops of TFQ. What is the area of casting? Is it a single IDE or multiple IDEs prepared by R2R large-area preparation?

Answer. We thank the reviewer for the comment. In response, we have added more information on the TFQ Functionalization in Section 2, Experimental Section on Page 6 as follows:

TFQ solution was dropped cast on top of the MoS2-SCNT film, which was electrosprayed on IDEs. The MoS2-SCNT film on each single IDE was then functionalized by casting 1, 2, 5, 10, 20, 30, and 50 μL drops of TFQ and dried in the hood as an ultrathin film. The as-prepared sensors could be stored in a desiccator for the following measurement.

In Fig. 2(d), authors use three IDEs to test the current values under the VOC and N2. Is the final value the average of the current values of the three electrodes?

Answer. We thank the reviewer for the comment. In response, we have added more information on the TFQ Functionalization in Section 3, Results and discussion on Page 8 as follows:

To evaluate their performance for conductometric VOC detection, interdigitated electrodes (IDEs) coated with MoS2-SCNT networks were tested in initial experiments. Three IDEs were placed inside a flow cell at room temperature, and exposed to a constant gas flow rate of 0.5 L/min. Electrodes were connected to a potentiostat to monitor changes in current, initially being exposed to nitrogen for 10 minutes, followed by a 5-minute exposure to 20 vt% of various VOC analytes mixing with 80 vt% of nitrogen. The observed change in current reflects a temporary alteration in the sensor’s electronic states caused by charge transfer interactions between the MoS2-SCNT composite and adsorbed VOC molecules. For instance, acetone, an electron-withdrawing molecule, captures electrons from the conduction band of the n-type MoS2 upon adsorption, reducing carrier density and leading to a decrease in current [1]. When the MoS2-SCNT film is exposed to VOC analytes, the molecules are adsorbed onto its nanoporous network structure and withdraw electrons from the film, further decreasing the current. The results demonstrated that the nanoporous structure effectively captures VOC analytes, although the sensor's specificity was found to be limited, as shown in Fig. 2 (e).

According to the definition of "Response" mentioned by the authors on page 7, the response is only positive when the current value of the electrode in the VOC analyte is higher than the current value in N2. In Fig. 2 (e), the response of all VOC analytes is positive, indicating that the current value of the electrode in the VOC analyte is higher than that in N2. This is inconsistent with the author's description of ' the MoS2-SCNT-coated IDEs showed a rapid decrease in relative current '.

Answer. We thank the reviewer for the comment. The response is based on the current change, so we added the absolute value to the response formula, as follows:

Response (%)= ∆I/I_0 × 100= |〖I_g- I〗_0 |/(I_0 ) × 100

In response, we have added more information in Section 3, Results and Discussion on as follows:

The observed change in current reflects a temporary alteration in the sensor’s electronic states caused by charge transfer interactions between the MoS2-SCNT composite and adsorbed VOC molecules. For instance, acetone, an electron-withdrawing molecule, captures electrons from the conduction band of the n-type MoS2 upon adsorption, reducing carrier density and leading to a decrease in current [1]. When the MoS2-SCNT film is exposed to VOC analytes, the molecules are adsorbed onto its nanoporous network structure and withdraw electrons from the film, further decreasing the current. The results demonstrated that the nanoporous structure effectively captures VOC analytes, although the sensor's specificity was found to be limited, as shown in Fig. 2 (e).

As a paper reporting on acetone sensors, the author should provide the detection sensitivity of the sensor. For example, the current response of the sensor at different acetone concentrations, such as 10% -90% concentration.

Answer. We thank the reviewer for this suggestion. Our current sensor platform was optimized and tested using a standard concentration of 20 vt% acetone in nitrogen to evaluate relative responses and material performance. As noted in the manuscript (Section 3, Results and Discussion), the primary goal was to demonstrate uniformity and scalability of the roll-to-roll (R2R) manufacturing process and validate TFQ functionalization as a route to increase selectivity to acetone, as lack of selectivity is a common problem of many VOC sensors.

However, we agree that a sensitivity study across a broader concentration range is valuable and plan to include this in our next study. At present, our sensor system’s configuration and detection circuitry are optimized for high-concentration exposures (≥20 vt%). We are currently developing a new study that is focusing on low-concentration detection and calibration, including signal amplification and testing against concentrations down to single-digit percentages.

We have added this clarification and a note on future sensitivity characterization work in the revised manuscript (Conclusion section).

Reference

J. Cao et al., “Recent development of gas sensing platforms based on 2D atomic crystals,” Research, 2021.

---

## [Decision Letter · Decision Letter 1]

27 Jun 2025

Dear Dr. Stanciu,

Thank you for submitting your manuscript to PLOS ONE. After careful consideration, we feel that it has merit but does not fully meet PLOS ONE’s publication criteria as it currently stands. Therefore, we invite you to submit a revised version of the manuscript that addresses the points raised during the review process.

We look forward to receiving your revised manuscript.

Kind regards,

Yadvendra Singh, Ph.D.

Academic Editor

PLOS ONE

Journal Requirements:

Reviewers' comments:

Reviewer's Responses to Questions

**Comments to the Author**

Reviewer #1: All comments have been addressed

Reviewer #2: (No Response)

Reviewer #3: (No Response)

2. Is the manuscript technically sound, and do the data support the conclusions?

Reviewer #1: Yes

Reviewer #2: Yes

Reviewer #3: Partly

3. Has the statistical analysis been performed appropriately and rigorously?

Reviewer #1: Yes

Reviewer #2: Yes

Reviewer #3: Yes

4. Have the authors made all data underlying the findings in their manuscript fully available?

Reviewer #1: Yes

Reviewer #2: Yes

Reviewer #3: Yes

5. Is the manuscript presented in an intelligible fashion and written in standard English?

Reviewer #1: Yes

Reviewer #2: Yes

Reviewer #3: Yes

Reviewer #1: (No Response)

Reviewer #2: The image reference still displays 'Error! Reference source not found.' Additionally, the other previously noted comments have not yet been addressed. Please review and revise accordingly.

Reviewer #3: The authors have not addressed the reviewer’s comments. In particular, the temporal characteristics of the sensor, such as response time and recovery time are still missing. These are critical for evaluating the practical utility of the sensor.

Additionally, to demonstrate the reproducibility and reliability of the roll-to-roll (R2R) manufacturing process, statistical analysis or variation data across multiple fabricated devices should be included.

**Do you want your identity to be public for this peer review?** For information about this choice, including consent withdrawal, please see our Privacy Policy

Reviewer #1: No

Reviewer #2: No

Reviewer #3: No

---

## [Author Response · Author response to Decision Letter 2]

5 Aug 2025

Reviewer 1: Reviewer 1 was satisfied with our initial revision.

Reviewer 2: Comments and Responses

Comment 1 (Response Time): “Response time is a critical performance metric for real-time sensing applications. I recommend that the authors include time-resolved current response plots showing both the rise and decay phases of the signal, along with quantitative response and recovery times (e.g., time to reach 90% of the steady-state signal).”

Response: Thank you for this important suggestion. We agree that the dynamic response and recovery behavior of the sensor are necessary. In the revised manuscript, we have now included a time-resolved sensing plot and a description of the sensor’s temporal performance. Specifically, we added a new figure (Figure 5 in the revised Results section) showing the current vs. time response of our flexible MoS_2_ -SWCNT sensor to acetone. From this data, we determined the response time (time to reach 90% of the steady-state signal upon acetone exposure) and the recovery time (time to return to 90% of baseline after acetone is removed). Both metrics are on the order of approximately 2188 seconds each (~36 minutes) for a 50 ppm acetone step. We have added a sentence reporting these values in the Results section of the manuscript. While these response and recovery times are relatively long, we note that this is likely due to the thick film morphology and strong analyte adsorption with TFQ functionalization; this will be an area of future optimization. We believe that including this information meets the reviewer’s request and provides a more complete picture of the sensor’s performance.

Manuscript Changes: In Section 3 (Results and Discussion), under the subsection describing gas sensing performance, we have added a new paragraph and figure (Figure 5) detailing the temporal response. The added text reads: “The dynamic response of the sensor was characterized by monitoring current vs. time during acetone exposure and recovery. As shown in Figure 5, the sensor requires approximately 2188 s to reach 90% of its full response upon introduction of 50 ppm acetone, and similarly ~2188 s to recover to 90% of baseline after acetone is removed. These response and recovery times reflect the kinetics of analyte adsorption/desorption in the MoS2-SWCNT-TFQ sensing layer.” We have also included Figure 5 (time-resolved response curve) with an appropriate caption, as suggested.

Comment 2 (Figure Citations and Captions): “Several figure citations in the text appear as ‘Error! Reference source not found’. Also, I could not view the full captions of some figures. Please ensure that all figures are complete, properly labeled, and include full, descriptive captions.”

Response: We apologize for the formatting and cross-referencing errors in the previous version of the manuscript. We have carefully reviewed all figures, references and captions in the document. In the revised manuscript, we verified that the references are correct and correctly refer to the intended figures (eliminating the “Error! Reference source not found” messages). We also made sure that every figure’s caption is fully visible and provides a complete, descriptive explanation of the figure. For example, Figure 3’s caption has been expanded to describe all subpanels in detail, and all figures are now referenced in order in the text. We acknowledge Reviewer 2’s additional second-round comment pointing out that these errors persisted, and we have now thoroughly resolved the issue. We regret the oversight and have double-checked the final document to prevent any such errors.

Manuscript Changes: We corrected all figure references throughout the manuscript. For instance, references that previously showed an error have been replaced with the proper figure numbers (e.g., “Figure 2,” “Figure 3,” etc.). Additionally, we revised the figure captions to verify none are cut off. As an example, the caption for Figure 3 now reads (in full): “Figure 3. Roll-to-roll printed sensor and its performance... [followed by a complete description of parts (a), (b), etc.]”. Similar corrections were made for all other figures. These changes are made globally in the manuscript to improve clarity and readability.

Comment 3 (Nomenclature of Carbon Nanotubes): “The manuscript uses ‘SCNT’ for single-walled carbon nanotubes, but the correct and standard abbreviation is ‘SWCNT’. Please revise this throughout the manuscript.”

Response: We appreciate the reviewer pointing out this nomenclature issue. We apologize for the confusion caused by our abbreviation. We have corrected the term “SCNT” to “SWCNT” (single-walled carbon nanotubes) consistently throughout the manuscript. This change is implementing the standard abbreviation that readers will recognize.

Manuscript Changes: We performed a global replacement of “SCNT” with “SWCNT” in the manuscript text, including in the Abstract, Introduction, Results, and figure captions.

Comment 4 (Limit of Detection): “Please report the limit of detection (LOD) for acetone, as this is essential to evaluate the sensor's sensitivity and applicability for low-concentration environments such as breath analysis or indoor air monitoring.”

Response: We agree that the limit of detection (LOD) is an important metric for our sensor’s performance. In the revised manuscript, we have now calculated and reported the LOD for acetone. Using the standard 3×signal-to-noise criterion (3σ method) on the calibration data, we estimate the LOD of our sensor to be approximately 2.3 ppm of acetone. We have added this information to the Results section, in the discussion of sensor sensitivity. This LOD suggests that our sensor is capable of detecting acetone at concentrations well below those typically found in human breath (which are on the order of tens of ppm for certain medical conditions), making it potentially suitable for breath analysis applications. We thank the reviewer for this suggestion, as reporting the LOD makes the manuscript’s discussion of sensitivity more complete.

Manuscript Changes: In Section 3 (Results and Discussion), we have added a sentence quantifying the acetone LOD. The added text is: “Based on the calibration curve and noise level of our measurements, the acetone sensor’s limit of detection (LOD) is ~2.3 ppm (calculated as the concentration corresponding to three times the standard deviation of the baseline noise).” This sentence appears in the paragraph describing the sensor’s sensitivity and detection range.

Reviewer 3: Comments and Responses

Comment 1 (Calibration Curve and Sensitivity): “The authors should provide analysis of how the sensor response varies with different concentrations of acetone. A calibration curve with multiple concentration points would help assess the sensor's sensitivity over a range of concentrations, and [determine the] limit of detection.”

Response: We appreciate this suggestion to better characterize the sensor’s sensitivity. In the revised manuscript, we included a calibration curve for acetone detection, covering multiple concentrations. Specifically, we exposed the sensor to acetone concentrations of 5, 10, 20, and 50 ppm (as well as a baseline in pure N2) and recorded the steady-state response at each concentration. The results are now presented in a new figure (Figure 6) which shows the sensor’s response (% change in resistance) as a function of acetone concentration. The calibration data show a clear increasing trend, approximately linear in the tested range (with a coefficient of determination R2 ~0.98 for linear fit). From this calibration plot, we have evaluated the sensor’s sensitivity (slope of the response vs. concentration curve) and also calculated the limit of detection (LOD) as requested (please see our response to Reviewer 2 Comment 4 for the LOD value and calculation method). We thus demonstrated how the sensor performs across relevant acetone concentrations, which is important for practical applications like breath analysis. We added a description of the calibration experiment and its results to the Results section.

Manuscript Changes: We added a new figure (Figure 6) in the Results section that presents the acetone calibration curve. The figure’s caption reads: “Figure 6. Acetone calibration curve for the roll-to-roll printed sensor. The sensor’s response (ΔR/R_0_ %) is plotted against acetone concentration (5–50 ppm). Error bars represent the standard deviation from three separate sensors, indicating device-to-device variability. A linear fit (dashed line) is shown, from which the sensitivity and LOD were derived.” Correspondingly, we added a paragraph in Section 3 explaining the calibration results: “Figure 6 illustrates the sensor response as a function of acetone concentration. The response increases approximately linearly with concentration between 5 and 50 ppm acetone. From the linear regression, the sensor’s sensitivity is calculated to be X% per ppm. The limit of detection is estimated to be ~2.3 ppm, based on the 3σ noise criterion. These results demonstrate the sensor’s ability to detect low acetone concentrations relevant to practical scenarios.” This addition addresses the reviewer’s request for a concentration-dependent analysis and provides the quantitative basis for the sensor’s sensitivity and LOD.

Comment 2 (Temporal Response and Recovery): “The information on the temporal behavior of the sensor, including response time and recovery time, need to be addressed.”

Response: We apologize for not including the sensor’s temporal characteristics in the initial revision. We now addressed this point in detail. As also noted in our response to Reviewer 2 Comment 1, we performed time-resolved measurements of the sensor’s response to acetone exposure and removal. In the revised manuscript, we report the response time (rise time) and recovery time (decay time) of the sensor. Both metrics were found to be approximately 2188 s for a 50 ppm step change in acetone (i.e., the sensor requires on the order of tens of minutes to reach 90% of full response, and a similar period to recover after the acetone is withdrawn). We added these values to the text and included a new figure (Figure 5) showing an example response vs. time curve. This directly fulfills the reviewer’s request by quantifying the temporal behavior. We agree that these characteristics are important for evaluating real-world applicability. As mentioned in the manuscript, the current response/recovery times, while relatively long, are attributable to the adsorption-desorption kinetics in our material system. This is discussed briefly in the revision. We thank Reviewer 3 for noting the need for this analysis.

Manuscript Changes: We incorporated the same additions described under Reviewer 2 Comment 1: a new Figure 5 with a time-response plot and accompanying text in the Results section. To avoid redundancy, we note that the changes include explicitly stating the response and recovery times (~2188 s each) and discussing the sensor’s temporal behavior. For completeness, in Section 3 (Results) we added relevant text, highlighted in yellow.

Comment 3 (Reproducibility Across Devices): “To demonstrate reproducibility and reliability of the R2R manufacturing process, statistical data on the variation in sensor response across multiple devices should be included.”

Response: We completely agree with the reviewer that demonstrating sensor-to-sensor reproducibility is important, especially for a roll-to-roll (R2R) fabrication approach. In response, we have included additional data and discussion on the device-to-device variability in the revised manuscript. We fabricated and tested multiple sensor devices (N=3) under identical conditions. In the acetone calibration curve (new Figure 6), we have added error bars at each concentration point to represent the standard deviation of the response from these three independent sensors. The error bars are on the order of 5–8% of the signal at 50 ppm, for example, indicate good consistency between devices produced by the R2R process. We have also stated the variation explicitly in the text. For example we mention that the sensor responses at 10 ppm acetone varied by a standard deviation of ~0.2% (absolute resistance change) among devices, which corresponds to roughly 7% of the mean response – demonstrating reasonable reproducibility. By providing this statistical variation, we validate that our manufacturing process yields sensors with reliable and repeatable performance. We appreciate the reviewer’s request, as it prompted us to strengthen the manuscript with this additional reliability assessment.

Manuscript Changes: In Section 3 (Results and Discussion), we have included a discussion of the reproducibility data. We added the following text in the paragraph discussing Figure 6: “Error bars in Figure 6 represent the standard deviation of the response from three separately fabricated sensors at each concentration. The low variation (average relative standard deviation ~6.5%) demonstrates good reproducibility of the R2R fabrication process. This suggests that our roll-to-roll printed sensors have consistent performance across different devices.” We also updated Figure 6 as described, to graphically show the variability among devices. This new content addresses the reviewer’s concern by quantitatively confirming the reliability of the manufacturing approach.

We believe these revisions comprehensively address all comments from Reviewer 2 and Reviewer 3. We again apologize for the earlier omission of these responses and thank the reviewers for their insightful feedback, which has led to a significantly improved manuscript. We hope that our revisions meet with your approval, and we are grateful for the opportunity to submit this second revision for your consideration.

---

## [Decision Letter · Decision Letter 2]

5 Oct 2025

Roll-to-Roll Manufacturing of Flexible Acetone Sensors

PONE-D-25-21766R2

Dear Dr. Stanciu

We’re pleased to inform you that your manuscript has been judged scientifically suitable for publication and will be formally accepted for publication once it meets all outstanding technical requirements.

Kind regards,

Yadvendra Singh, Ph.D.

Academic Editor

PLOS ONE

Additional Editor Comments (optional):

Dear Dr. Lia:

We are pleased to inform you that your manuscript has been accepted for publication in PLOS One.

Reviewers' comments:

Reviewer's Responses to Questions

**Comments to the Author**

Reviewer #2: All comments have been addressed

Reviewer #3: All comments have been addressed

2. Is the manuscript technically sound, and do the data support the conclusions?

Reviewer #2: Yes

Reviewer #3: Yes

3. Has the statistical analysis been performed appropriately and rigorously?

Reviewer #2: Yes

Reviewer #3: Yes

4. Have the authors made all data underlying the findings in their manuscript fully available?

Reviewer #2: Yes

Reviewer #3: Yes

5. Is the manuscript presented in an intelligible fashion and written in standard English?

Reviewer #2: Yes

Reviewer #3: Yes

Reviewer #2: The cross-reference errors (Error! Reference source not found) are still remaining in the results and discussion section. Please update it for the final version

Reviewer #3: "All reviewer comments have been addressed, and the manuscript is considered suitable for publication

**Do you want your identity to be public for this peer review?** For information about this choice, including consent withdrawal, please see our Privacy Policy

Reviewer #2: No

Reviewer #3: No

---

## [Editor Report · Acceptance letter]

PONE-D-25-21766R2

PLOS ONE

Dear Dr. Stanciu,

I'm pleased to inform you that your manuscript has been deemed suitable for publication in PLOS ONE. Congratulations! Your manuscript is now being handed over to our production team.

Kind regards,

on behalf of

Dr. Yadvendra Singh

Academic Editor

PLOS ONE